# The Influence of Muslim and Christian Destinations on Tourists’ Behavioural Intentions and Risk Perceptions

**DOI:** 10.3390/bs14040347

**Published:** 2024-04-22

**Authors:** Rita R. Carballo, Carmelo J. León, María M. Carballo

**Affiliations:** University Institute of Tourism and Sustainable Economic Development—TIDES, University of Las Palmas de Gran Canaria, 35017 Las Palmas de Gran Canaria, Spain; carmelo.leon@ulpgc.es (C.J.L.); magdalena.carballo@ulpgc.es (M.M.C.)

**Keywords:** risk perception, Muslim, tourism, image, behavioural intentions, multi-group analysis

## Abstract

This paper studies the factors that influence tourists’ risk perceptions of various destinations with different attributes and sociocultural profiles. Factor analysis is utilised to investigate the determinants of risk perceptions, finding that they are influenced both by the type of risk (delinquency, health, accident, environment and catastrophe) and by the characteristics of the destination regarding the management of risk. Structural equations modelling is conducted to study the relationships between risk perceptions, destination image and visiting intentions across destinations. Multi-group analysis across different destinations proves that tourists’ risk perceptions have different influences on destination image and visiting intentions. The results show that there are significant differences according to the predominant religion at the destinations, i.e., Muslin and Christian. The implication is that different dimensions of perceived risks and destination socio-culture contexts have different influences on the behaviour of tourists.

## 1. Introduction

Tourist safety at a destination is one of the most essential conditions that characterise the quality of experience [1]. The economic development of any country depends mainly on its security [2]. Especially in tourism, it is a major factor in tourism promotion and demand. In addition, security is a basic attribute for a tourist destination to be competitive, and no marketing strategy will be successful because of the lack of political stability and security [3]. Therefore, it is important to project an image of a safe destination in the marketing campaigns of tourist destinations.

Risks to personal security can be both perceived and unperceived. Perceived risk commonly influences the tourist’s choice, even if it is not real [4]. Nevertheless, the unperceived risk will not alter the tourist’s decision-making even if it were real [5,6]. In tourism, risk perception is inherent to consumers’ decision-making. The understanding of tourists’ perceptions about security in a destination is important for creating a favourable environment for tourism development.

Risk is a determining factor in a tourist´s choice of destination, the assessment of alternative destinations, and the decision on the type and amount of expenditure in the chosen destination [7,8]. Further, the perceptions of risks have been shown to vary according to both tourists’ and destinations’ characteristics [1]. Thus, it is desirable to study and identify the elements or attributes that influence risk perception.

There are different types of risks that tourists face, which can damage both the image of a destination and the travel intentions of tourists. Common risks have been found to influence both tourists and locals at a destination, like crime, health or natural disasters [9]. However, there are other specific risks related to the tourist consumption process, mainly those related to the transport chosen for the journey, the hostility of residents, bad weather conditions, cultural barriers, lack of hygiene that may be encountered at the destination, the uncertainty arising from the regulations and laws in the countries of destination [10]. Nevertheless, the risks that are most relevant to tourists’ choices commonly arise from the lack of personal security, common delinquency, theft and fraud [11,12].

Even though there is a large variety of risks to be considered, risks can also vary according to the context in which they are [13]. For instance, social and cultural contexts can influence the risk perception of tourists [14,15]. However, the influence of cultural contexts underpinned by religious practices and values has not been given much attention in earlier research [16]. When it comes to tourism, most research has been concerned with the religion of the tourist rather than the specific religious attributes of the destinations [17,18]. 

The objective of this study is to assess the influence of the predominant religion in tourist destinations on the risk perceptions of tourists and their behavioural intentions. The predominant religion of a tourist destination conditions a set of cultural values and social customs that may potentially influence the perceptions of tourists [19]. Thus, this study tests the hypothesis of whether tourists’ risk perceptions and their influence on behavioural intentions and destination image are different across destinations with different religions. To this aim, the risks at tourist destinations that are assessed by tourists are grouped into those in which the most predominant religion is Muslim and those in which is a Christian religion. Since the potential risks affecting tourists’ perceptions and their choice of destination can be wide-ranging, we consider the influence of various types of risks, such as health risks, the risks of delinquency, accident risks, environmental risks and risks of disasters, both on the image of destinations, including cognitive and affective factors, and on the behavioural intentions.

In order to study the moderating influence of cultural religion at destinations, a multi-group analysis is conducted by grouping the studied countries by those aligning with one of the predominant religious beliefs. The results highlight the influence of the predominant religion at the destinations on the risk perceptions and behavioural intentions of tourists. From a theoretical perspective, the present research points out the need to take into account religious contexts on the formation of risk perceptions, thereby drawing on some of the postulates of cultural theory [20]; as a practical implication, the management of risks at tourist destinations should take account for the influence of religious culture on the perceptions of tourists. 

## 2. Literature Review

Risk perception has been defined in many different ways in tourism research, as the potential to lose something of value [21] or as the sum of negative outcomes and the probability of their occurrence [2].

The risk perception is subjective; it can be considerably different depending on the type of risk [22]. It is crucial to understand the risk formation process by tourists because it will influence their decision-making. Consequently, it will help to establish strategies to improve the destination image. In this review, we first consider the influence of risk perception on a destination’s image, and then we focus on reviewing the ideas regarding the influence of risk perception on visiting intentions.

Central to the perception of risks at a destination are those cultural and socioeconomic characteristics that revolve around religious values and customs [23]. Along this line, various cross-cultural studies have found that common cultural values may raise common fears and other emotions, thereby deterring tourists from traveling to some destinations [23]. Some studies have found differences in risk perceptions across individuals with different religious beliefs, i.e., Catholics, Protestants and Jews [24].

In the area of travel and tourism, the study of [17] applied regression analysis to prove the relationship between religiosity and perceived travel risks, as well as multivariate analysis of variance to assess potential differences in risk perceptions across religious affiliations. It was found that religiosity, religious affiliation and perceived risk dimensions discriminated among preferred travel styles for a future trip [17].

Thus, it is important to consider, in a much broader context that involves cultural and religious frames, what the determinants of perceived risk are. More generally, risk perceptions are elements of destination competitiveness that influence the decisions tourists make regarding the choice of destinations and the consumption process once at a destination [3,25].

### 2.1. Influence of Risk Perception on the Image of a Destination

The image of a destination is the sum of beliefs, ideas and impressions that one person has about a destination [2]. There have been developments in the literature about the influence that cognitive (functional) and affective (emotional) components have on the perception of a tourist destination’s image [21]. The cognitive component is related to those material aspects that have a role in the tourist services provided, such as the quality of infrastructures, the natural environments or the public services. The emotional image is based on the emotions that arise in tourists at the prospect of visiting a destination.

It is well recognised that personal security and safety are important factors that influence the attractiveness of tourist destinations. Ref. [26] show that the perceived risk can affect tourist satisfaction through destination image. Moreover, Ref. [27] find differences in the impact of risk perception on destination image for first-time visitors and repeat visitors. Ref. [28] argue that perceived risk influences image in different ways depending on the information sources.

Tourists are definitely attracted by good impressions, and the existence of security and safety is essential for it [29]. Ref. [30] point out that tourists seek destinations that are familiar and close to them when travelling internationally. In order to be a successful destination, there needs to be an adequate level of security and safety. The tourism sector is highly dependent on security [31]. Therefore, security is a dimension integrated into the perceived image of a destination, but the relevance of each dimension of perceived risks in a destination’s image has not been tested [30]. There can be different dimensions of security that have different ways of influencing destination image. In addition, the image of a destination is also based on cultural aspects, which can include those related to religious culture and values [32,33].

Thus, based on the literature review presented above, there is a need for further research on the relationships between risk perception and the image of the destination that focuses on the potential role of the predominant religion at destinations as a moderator of this relationship. Thus, the following hypotheses and sub-hypotheses are proposed, which are tested later with the empirical investigation:

**H_1_.** *Risk perception significantly and negatively affects the image of the destination*.

**H_1a_.** *The predominant religion at destinations has a moderating effect on the relationship between the perceived risk and the image of the destination*.

### 2.2. Influence of Risk Perception on Behavioural Intentions

Empirical evidence shows that the level of risk perception has an influence on destination choice, partly because of potential tourists’ high reliance on external information rather than their own experience. The media has also been found to have an important role in exacerbating security-related issues at destinations influencing travel flows [34,35].

Thus, the risk influences tourists’ choices. Essentially, it can be stated that there is a significant and positive relationship between the level of risk perception concerning a destination and a tourist’s willingness to travel to the same destination. Ref. [36] found that a majority of tourists tend to change their travel plan when they face a high risk at a destination. People are rarely willing to change their place of residence but are very sensitive to risks when deciding upon their travel destination choice since there is nothing that can force them to choose a destination they perceive as insecure [3].

Furthermore, there are also studies reporting that threats to safety and security have caused a decrease in tourist demand at a destination. There is also a common substitution effect observed in tourists’ choices between destinations when terrorism or political turmoil strikes [37], while the reactions of tourists to terrorism may be different depending on their nationality and previous experience [38]. Further, it is very likely that the perception of risk will eventually affect loyalty to the destination and the recommendation to visit it [39]. Therefore, the analysis of the impact of risk perception should be based on an exploration of how the specific risks and risk dimensions affect the different types of tourists and the decision to travel to a destination. It can be ascertained that the different dimensions of risk perception can have important effects on tourists’ decisions.

To the authors’ knowledge, there are no available studies that analyse how predominant religious beliefs of destinations may impact tourists’ visits or behavioural intentions. Hence, this paper proposes the following additional hypotheses:

**H_2_.** 
*Risk perceptions significantly and negatively affect behavioural intentions.*


**H_2a_.** *The predominant religion at destinations has a moderating effect on the relationship between perceived risk and behavioral intentions*.

### 2.3. Influence of Image Destination on the Intentions to Visit

Since tourists have very little knowledge of destinations they have not yet travelled to, and since tourists frequently perceive the image according to their experience visiting the destination, the literature corroborates that the image and knowledge of a destination have an influence on the decision to visit it [40]. An image can help tourists pertest to the destination, thereby influencing the choice process. Thus, it is clear that destination image influences how tourists decide about a destination, the subsequent evaluation of the vacation and their future intentions to visit. Some of the factors of the destination image are founded on cultural aspects to which religious values and customs can be assimilated [32]. Studies show that there is a positive relationship between the image of a destination and the intention to visit [11,16]. Image management can also be utilised as a part of a useful strategy to influence tourists’ decision-making since it plays a central role in tourist behaviour [41,42,43]. Further, the relationship between image and behavioural intentions can also be affected by the predominant religion at tourist destinations. Thus, the following hypotheses are proposed:

**H_3_.** *The image of a destination significantly and positively alters behavioural intentions*.

**H_3a._** *Predominant religion at destinations has a moderating effect on the relationship between the image of the destination and behavioural intentions*.

Figure 1 illustrates the relationships that these hypotheses try to evaluate with the empirical investigation presented in the next section.

## 3. Research Methods

### 3.1. Study Site and Sampling

In order to test the hypotheses outlined above, an empirical study was proposed to investigate the causal relationships between perceived risk, image and tourist behaviour of tourists of Germany and the United Kingdom, the major tourism source markets in Europe [44]. The destinations under investigation included Muslim destinations (Egypt, Morocco and Indonesia) and Christian destinations (Spain, Brazil and Colombia). Also, the aim of analysing these destinations was due to their different levels of economic and tourism development, a developed tourist destination (Spain) and some emerging regional destinations with different perceptions regarding their security levels. Furthermore, there are differences in terms of the distance from the countries of origin to the destinations (long and short haul) as well as the religious and cultural values of the destinations. In Egypt, Morocco and Indonesia, the predominant religion of the population is Muslim (78.61%, 95.18%, and 84.35%, respectively), while in Spain, Brazil and Colombia, it is Christian (66%, 90% and 93%, respectively). Surveys were completed during the months of June and August of 2023 through web interviewing by specialising survey companies working throughout Europe. The questionnaires were tested in the United Kingdom and Germany. A total of 1212 questionnaires were answered: 603 in the United Kingdom and 609 in Germany. A random sampling method was applied with quotas for gender and age to ensure the sample was representative. Table 1 shows the summary sample statistics for the socioeconomic variables.

The surveys were based on a thorough literature review on the topics of image evaluation and risk perception to test the hypotheses on the relationships between destination image, risk perception and visit intentions. Thus, there were questions about these specific variables. The image of the destination was evaluated utilising various questions focusing on the different dimensions of an image, similar to other studies measuring the destination image [45,46]. A group of 27 items focused on analysing the cognitive images of each destination on a seven-point Likert-type scale (1 = Total disagree; 7 = Total agree). To measure the affective item of the destination image, a 6-item, 7-point bipolar semantic differential scale was used [47]. In order to ascertain the intention to visit the destinations, a question was focused on the intention of traveling in a short time to each of the destinations using a 7-point Likert-type (1 = Very unlikely; 7 = Very likely).

Risk perception at each destination was estimated in two stages. The first stage took into value 25 items (see Table 2) related to the probability of suffering different types of risks at the destinations using a 7-point Likert-type response format (1 = Total disagree; 7 = Total agree). The second stage considered some risk management policies at the destination that make visitors perceive destinations as more risky; for this approach, 4 characteristics of the destination were evaluated (not existence of reliable health care service, non-security alarms, not communicating in your language and no police assistance) using a 7-point Likert-type scale (1 = Very unimportant; 7 = Very important). The last part of the survey contained the socio-demographic characteristics of individuals.

### 3.2. Modelling

An Exploratory Factorial Analysis (EFA) was used to determine the underlying dimensions of risk perception and the cognitive and affective images of the destination. Then, a Confirmatory Factorial Analysis (CFA) allowed for checking the convergent validity of the EFA scales.

Then, utilising the results from the CFA, a Structural Equations Model (SEM) was conducted with the aim of proving the hypotheses regarding the relationships between perceived risks, destination image and visit intentions. All data in the two stages were processed utilising SPSS 27 and AMOS 27. Finally, a multi-group analysis was carried out to determine the moderating effects of the predominant religion at the destinations on the causal relationship among risk perception, destination image and tourist behaviour.

## 4. Results

### 4.1. Measurement Model

#### 4.1.1. Factorial Analysis of Risk Perception

There were 25 questions regarding possible risks at tourist destinations that were questioned to tourists. The results of the EFA for the perceived risk items led to five dimensions (health risk, risk of suffering delinquency, accident risk, environmental risk and risk of disasters) (see Table A1). The KMO measure of 0.963 and Bartlett’s test of sphericity (X^2^ = 1.151, df = 411; *p* < 0.000) show that the data are feasible for the factorisation since the relative value of X^2^ with respect to the degrees of freedom (X^2^/gl) should not exceed 3 [48]. The five factors explained a total variance of 72.58% and Cronbach’s alpha values ranging from a low of 0.741 for factor 5 to a high of 0.896 for factor 2, showing that the factors are reliable.

The measurement models of the latent variables (risk perception and cognitive image of the destination) were confirmed using CFA to test the composite reliability, convergent and discriminant validity. Constructor validity was achieved since the Fitness Indexes exceeded the suggested threshold (>0.9) (GFI = 0.814; AGFI = 0.776; CFI = 0.895; TLI = 0.883; IFI = 0.896; RFI = 0.873; NFI = 0.886). The general fit of the model is acceptable since the X^2^ statistic of 384 (df = 142, *p* < 0.000) was significant, and the ratio of X^2^/df was 2.7, considered acceptable. The results indicate that convergent validity was accomplished since values for composite reliability (CR) and average variances extracted (AVE) are greater than 0.5 [49]. Discriminant validity was proved by showing that the squared correlations between a pair of constructs do not exceed their AVE estimates [50], which the analysis confirms.

Each risk attribute shows a different weight in each factor (Table 2). The risk of an illness dimension is explained because the health risk of an illness being transmitted by some animal (β = 0.83), which is the most important attribute, followed by the risk of finding a poor health care system at the destination when it is needed (β = 0.82). However, the risk of contracting some sexually transmitted disease (β = 0.72) is the attribute with the lowest importance, followed by the risk of getting ill because of consuming some food or drink in a bad state (β = 0.80). Therefore, the risk attributes that can be controlled by tourists are the ones with the lowest relative importance, whereas those that are perceived as uncontrollable are the ones most relevant to tourists.

The risk of being subject to delinquency is explained by the risk of being kidnapped (β = 0.87) and being a victim of a rape or sexual attack (β = 0.86), which have larger impacts than the risk of being robbed with physical violence (β = 0.73) and the risk of a terrorist attack (β = 0.67). These differences can be explained by the different probabilities of occurrence of these risks and their impacts.

With respect to the factor of risk of accidents, this is explained by the risk of fire at the hotel (β = 0.86) and the risk of an air accident (β = 0.79), followed by the risk of a traffic accident (β = 0.77) and the risk of drowning in the sea or in a swimming pool (β = 0.73). Thus, it seems that tourists’ perceptions of accident risks are influenced by the media and the news rather than by objective factors.

The environmental risk is explained because of the risk of encountering adverse weather conditions (β = 0.79) and high levels of noise (β = 0.77), followed by the risk of exposing themselves to high solar radiation (β = 0.68). Again, those uncontrollable risks are perceived as more relevant. With regard to the risk of disasters, we find that the risk of natural disasters is much more relevant (β = 0.90) than the risk of disaster caused by human factors (β = 0.76).

#### 4.1.2. Factorial Analysis of Destination Image

An image of a destination has been defined as based on two different aspects: cognitive and affective. The results of the EFA to the six affective items of destination image (healthy, active, sustainable, authentic, pleasant and exciting destination) consisted of one factor named “affective image”.

The application of EFA to the twenty-seven cognitive image of the destination consisted of three items (IC1, IC2 and IC3) (see Table A2) which confirmed the suitability of the data for factorisation with KMO of 0.969 and Bartlett’s test of sphericity (X^2^ = 982, df = 351, *p* < 0.000). The three factors explain 68.97% of the total variance and are moderately reliable, with Alpha Cronbach ranging from 0.888 to 0.949. The CFA of the destination image (Table 3) shows that constructor validity was achieved (>0.9) (GFI = 0.978; AGFI = 0.965; CFI = 0.962; TLI = 0.950; IFI = 0.979; RFI = 0.953; NFI = 0.983). Chi-square’s ratio was significant (X^2^/df = 2.2). Convergent validity and discriminant validity were confirmed.

### 4.2. Structural Equations Modelling

The SEM approach makes a simultaneous estimation of the relationships between the latent variables and the attributes, or observed variables, and the determination of the validity and reliability of the measures [51], We apply the SEM approach to the relationships between the perceptions of security at the destination, the destination image and the intentions to visit, allowing us to investigate the hypotheses concerning the nature of these relationships.

The results of the structural model (Figure 2) show coefficients in a standardised form. The results of the SEM prove that the goodness of fit is satisfactory (X^2^ = 782.1, gl = 266; *p* < 0.00). The estimated indices are above 0.9 (GFI = 0.930; AGFI = 0.997; CFI = 0.956; TLI = 0.943; IFI = 0.956; RFI = 0.909; NFI = 0.930) and the RMSEA index is below 0.05, therefore under the acceptable limit.

The regression coefficients are all significant since the statistics are below 0.05 level (5%). These suggest that the hypotheses are proven with the empirical data, thereby accepting the three hypotheses. Thus, the perceived risk through its different dimensions has a significant and negative impact on the image of the destination and on the intentions to visit (β = −0.46 and β = −0.51; *p* < 0.000), thereby accepting H_1_ and H_2_ (see Table 4). This means that the higher the risk perceived by visitors, the higher the negative image of the destination. Therefore, the higher the risk perceived by tourists, the greater the impact on their behaviour. We can also state that the image of the destination has a significant and positive impact on the intentions to visit the destination (β = 0.48; *p* < 0.000), thus accepting H_3_. Thus, the intention to visit a destination will be higher whenever it offers a better image. The perceived risk also has an indirect influence on the future intentions to visit the destination through the destination image.

In addition, the results of the study show that the perceived risk depends significantly on the type of risk and specific attributes of the destination. In relation to the types of risks, it can be noted that the perceived risk of the destination depends mainly on the health risk (β = 0.91), followed by the accident risk (β = 0.90), risk of suffering delinquency (β = 0.77), the environmental risk (β = 0.70), and the risk of disasters (β = 0.59). With respect to the destination risk management policies affecting risk perception, tourists are more impacted by the lack of security alarms in hotels (β = 0.78), the lack of police (β = 0.72) and the lack of accessible health services (β = 0.71), and much less important by not being served in their native language (β = 0.60).

### 4.3. Moderating Effects

This research also tested the moderating effects of predominant religion at destinations on each path proposed in the model. For the moderation tests, the sample was divided into two subgroups based on the destinations: Muslim destinations (Egypt, Morocco and Indonesia) and Christian destinations (Spain, Brazil and Colombia). The sizes were 606 for each group.

A multi-group approach was used to compare the existence of invariance across groups of informants [52]. The objective of the multi-group analysis is to examine how the relationships among risk perception, destination image and behavioural intentions vary across groups of the predominant religion at the destination.

The analysis was run by comparing the groups on the same specified model while constraining the parameters to be equal across groups (constrained model), thereby generating an overall chi-square value for the sets of sub-models as part of a single structural system. Next, no equality constraints specified across the groups (unconstrained model) were considered, resulting in a second chi-square value. A significant difference in the chi-square values between the two models determines the presence of moderator effects. That is, if the change in the chi-square values is statistically significant (the corresponding confidence level is 95%), the null hypothesis of parameter invariance is rejected, and a moderator effect is indicated [53]. The interpretation of these results yields a great deal of interesting information from both managerial and policy perspectives. Hence, it appears the predominant religion moderates the path relationships among perceived risk and destination image (ΔX^2^ = 8.5, *p* = 0.004). Each of the beta coefficients (Table 5) explained the relative importance of the risk perception in relation to the destination image. The coefficient of Muslim destinations (β = −0.31) was greater than that of Christian destinations (β = −0.17).

The finding suggests that the destination plays a moderating role in the relationship between perceiver risk and behaviour intentions (ΔX^2^ = 5.7, *p* = 0.017). The coefficient of Muslim destinations (β = −0.18) was greater than that of Christian destinations (β = −0.11). The results also present that the different groups moderate the path relationship between destination image and intention to visit (ΔX^2^ = 1.7, *p* > 0).

## 5. Discussion

Numerous studies about the behavioural consequences of risk environments have demonstrated the importance of risk perceptions for people’s decisions and actions [16]. Much of the research on risk perceptions to date analyse risk perceptions from a cross-cultural perspective. The cultural theory of risk is capable of “predict and explain what kind of people will perceive which potential hazards to be how dangerous” [13] based on cultural backgrounds. However, to the best of our knowledge, there is not much research about the perceived travel risks associated with the predominant religious affiliations across different destinations [17]. Most research about travel and tourism risk has been concerned with the characteristics of risk perceivers or the religion of the tourist rather than the specific religious attributes of the destinations [18].

The concept of risk perception has been highly studied in tourism; however, the literature remains fragmented, resulting in a lack of a cohesive and comprehensive framework [1]. Previous studies have found that tourists’ risk perception has a high probability of impacting the image of tourist destinations and the intention to visit a destination [36], but little or no evidence is available on the role of religion values or related cultural backgrounds.

The present research shows that risk perception is an important factor that can influence both destination image and behavioural intention [40]. Therefore, the results are aligned with other studies that confirm that risk perceptions are an important factor that can influence both the destination image and behavioural intentions [32]. That is, there is a negative and significant impact of the risk perceptions on the destination image. Thus, the higher the risk perception, the lower the destination image [2]. Also, there is a negative and significant impact of risk perception on the intention to visit a tourist destination, i.e., the higher the risk perception, the lower the intention to visit a destination [37]. Further, it is shown that these behavioural influences vary according to the type of country to be visited based on the predominant religion and related culture of the recipient society. Thus, travel and tourism risk perceptions and their behavioural implications are also affected by the predominant religion of the destinations and not only by the individual sociological and cultural factors that are considered in the cultural theory of risk perception [13].

Therefore, results confirm that risk perceptions are dependent on both the different dimensions of risks (health, accident, delinquency, environment and disasters) and the specific attributes of the destination concerning the management of risk (non-security alarms, no police presence, not existence of health care service and not communicating in your language) [9]. But, to these factors, the role played by the different religious and cultural backgrounds of the destinations, which may influence the configuration of the image of the destinations and the risk perceptions of tourists, should be added. These results are in line with and resonate with those of previous studies that have highlighted the role played by cultural environments on the image of tourist destinations and their behavioural intentions [10].

## 6. Conclusions

Risk perception is an important aspect of the management of tourist destinations. Since tourists are sensitive to risks and security, these factors are essential for providing quality tourist experiences and for the sustainable development of tourist destinations. In addition, destinations are not all alike in the way that they raise the perceptions of risks on tourists and potential visitors [54], with sociocultural factors such as religious beliefs and customs potentially influencing the formation of product images and perceptions.

In this paper, we investigated the perception of risks by tourists and its relationships with destination image and the intention to visit a destination, taking into consideration the potential differential effects caused by the type of country according to predominant religious beliefs [24]. The results confirmed the hypotheses that relate tourists’ risk perceptions, as explained by the different types of risks and the characteristics of risk management at the destinations, with the destination image and the intentions to visit.

In addition, the empirical results of this research also prove that there is a moderating effect of the predominant religion at the destinations on the relationships between risk perceptions and both behavioural intentions and destination image [17,18]. That is, the predominant religion at the destinations, i.e., whether Muslim or Christian, significantly influences the strength of the relationships between perceived risks, the image of the destination and the intentions to visit the destination. Thus, the implication is that there are indeed differences in the perceived risks and their relationships that vary according to the different countries based on religious beliefs and customs. The moderating effect also applies to the different dimensions of risk, so these dimensions have a different influence on the perceived risk according to the destination in question.

### 6.1. Theoretical and Managerial Implications

From a managerial perspective across all religious types of destinations, it is extremely important to minimise risk perception among tourists that destinations take risk prevention measures such as the implementation of security alarms in tourist facilities, the increase in police presence, and the provision of reliable and accessible health services. In general, it has been found that certain risks such as illness transmitted by an animal, being kidnapped, sexual assault, fire at a hotel, air accident and adverse weather conditions are perceived as more relevant characteristics at the destination, explaining the different types of risks than other types of risks such as sexually transmitted disease, food poisoning, drowning in water or exposure to solar radiation.

Therefore, the results of this paper suggest that both the destination image and the inflow of tourists to a destination can be considerably improved by working through the components that affect risk perception, such as the specific attributes of the destination managing risk and the type of risks that can be minimised by prevention and policy measures. In addition, there is a need to work through these aspects to tailor the risk perceptions of tourists to Muslim destinations rather than Christian destinations. Thus, the understanding of the components affecting tourist risk perception can be useful for managing destination image and making it more attractive to tourist demand.

### 6.2. Limitations and Prospect Directions

The methods and results shown in this paper are not without limitations. Unfortunately, surveys have a great limitation in capturing the complexities of the religious mind, and the results can be contradictory or misleading in some aspects [55]. This study indicates the potential effect of the predominant religion within the cultural context of the tourist destination; therefore, future research should be carried out, isolating the religious aspect from the cultural context. That is, there is a need to isolate the religious influence from other sociocultural factors that are embedded in tourist destinations. On the other hand, the cross-sectional nature of the study limits the capture of social changes that can only be evidenced with longitudinal data. Along this line, a more qualitative approach could help to understand the religious context that influences the risk perceptions and behavioural intentions of tourists. Finally, the predominant religions focused on in this study were Christian and Muslim; hence, for future research, it could be useful to have more evidence from other destinations and other religious settings.

## Figures and Tables

**Figure 1 behavsci-14-00347-f001:**
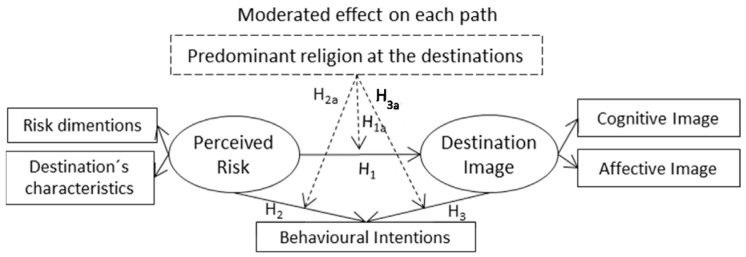
Proposed theoretical model.

**Figure 2 behavsci-14-00347-f002:**
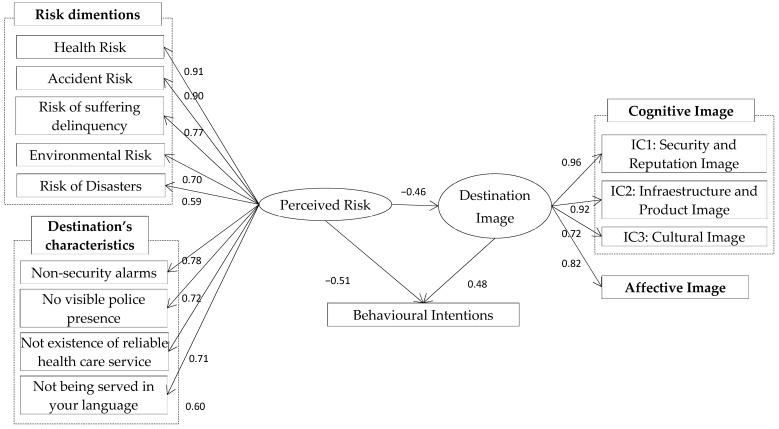
The estimated structural model.

**Table 1 behavsci-14-00347-t001:** Demographic data on sample.

Variables	United Kingdom	Germany
**Country of origin**	603	609
**Gender** %		
Male	49.2	51.3
Female	50.8	48.7
**Age** %		
16-24	17.9	18.7
25–34	17.6	18.5
35–44	21.3	21.1
45–54	21.2	21.7
55–64	12	11
More than 64	10	9
**Educational level** %		
Primary studies	11.3	12.4
High school	36.2	35.6
Low university degree	28.5	29.1
High university degree	24	22.9

**Table 2 behavsci-14-00347-t002:** Reliability and validity of the measurement items for risk perception.

Scale and Item	Standardised Loadings	Composite Reliability (CR)	Average Variance Extracted (AVE)
**Health risk**		0.872	0.623
Contracting vector-borne diseases (malaria, etc.)	0.83		
Deficit in health care if needed	0.82		
Contracting disease by ingesting food or drink	0.80		
Contracting a sexually transmitted disease	0.72		
**Risk of being subject to delinquency**		0.745	0.529
Being kidnapped	0.87		
Being a victim of rape or sexual assault	0.86		
Being assaulted with physical violence	0.73		
A terrorist attack			
**Accident risk**		0.868	0.622
Availability and effectiveness of fire safety measures and emergency service in the hotel	0.86		
Being involved in a plane crash	0.79		
Being involved in a traffic accident	0.77		
Drowning at sea or in a pool	0.73		
**Environmental risk and risk of disasters**		0.792	0.560
To have adverse weather conditions	0.79		
Experiencing high levels of noise	0.77		
Excessive sun exposure	0.68		
**Risk of disasters**		0.818	0.694
Natural disasters (earthquakes, storms, tsunamis, volcanic eruptions, forest fires, etc.)	0.90		
Human-induced disasters (nuclear accidents, toxic spills, etc.).	0.76		

Significant (*p* < 0.05).

**Table 3 behavsci-14-00347-t003:** Reliability and validity of the measurement items for cognitive image of destination.

Scale and Item	Standardised Loadings	Composite Reliability (CR)	Average Variance Extracted (AVE)
**IC1: Environment and Reputation Image**		0.933	0.700
Destination that offers personal security (few robberies, etc.)	0.87		
Destination with good overall quality of life for its residents	0.86		
Clean destination	0.85		
Destination with good environmental situation and pollution-free	0.83		
Destination with good reputation	0.81		
Family destination suitable for children	0.80		
**IC2: Infrastructure and Product Image**		0.942	0.729
Destination with a wide and varied range of sports (golf, tennis, water sports, etc.).	0.89		
Destination with good infrastructure of hotels, apartments and bungalows	0.88		
Destination with good nightlife (discotheques, pubs, etc.)	0.86		
Destination with wide and varied range of leisure activities (excursions, amusement parks, etc.).	0.86		
Destination with a good general infrastructure (roads, airport, transportation, etc.).	0.84		
Destination with extensive facilities for shopping	0.79		
**IC3: Cultural Image**		0.846	0.530
Destination with interesting customs and traditions	0.92		
Destination with historical–cultural places of interest to visit (museums, monuments, buildings, etc.)	0.75		
Destination with cultural activities of interest (festivals, concerts, folklore, etc.).	0.71		
Exotic destination	0.66		
Destination with great gastronomic variety and quality	0.55		

Significant (*p* < 0.05).

**Table 4 behavsci-14-00347-t004:** Structural path estimates.

Direct Effect	Estimates	*p*	Results
H_1_: Perceived Risk → Destination Image	−0.46	0.000	Supported
H_2_: Perceived Risk → Behavioural Intentions	−0.51	0.000	Supported
H_3_: Destination Image → Behavioural Intentions	0.48	0.000	Supported

Significant (*p* < 0.05).

**Table 5 behavsci-14-00347-t005:** Multi-group structural path estimates. Predominant religion at destination.

	Constrained Path(s) Model	Path Coefficient	
Path Relationships ML (95%) (1.212)	ΔX^2^	Muslim Destinations	Christian Destinations
All	33.6		
Perceived Risk → Destination Image	8.5	−0.31	−0.17
Perceived Risk → Behavioural Intentions	5.7	−0.18	−0.11
Destination Image → Behavioural Intentions	1.7	0.88	0.70

Significant (*p* < 0.05).

## Data Availability

Data are available from authors upon request.

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
