# Peer review of "The Influence of Muslim and Christian Destinations on Tourists’ Behavioural Intentions and Risk Perceptions"

_behavsci, 2024, doi:10.3390/bs14040347_

Round 1
Reviewer 1 Report
Comments and Suggestions for Authors
Dear authors I found this research very interesting. The idea of investigating religion in perceived risk for destination image to be useful for both practical and theoretical implications. I found your manuscript interesting and engaging however I have made some notes:
Introduction: Section has good flow easy to follow. It presents the topic of discussion and the need for the research. I believe it would benefit from having the aim of this paper more clearly presented in the end and some expected theoretical and practical implications of this research.
Literature review: Section has some very good arguments and adequate current relevant evidence to support them. However, it does not serve the purpose to define several important topics such as: perceived risk, unperceived risk, personal security, safety, terrorism, political turmoil strike. Furthermore, looking at the hypothesis chapter it does not define and discuss topics such as religion, cognitive image, affective image and general image.
Hypotheses: This section is a bit weak. It does state that “Based on the literature review presented above we formulate some hypotheses”, but it does not present any previous literature that examined the relationship between the proposed hypotheses.
Figure 1: Figure only presents H1, H2 and H3 ignoring H1a, H2a and H3a. It also introduces new topics (cognitive image, affective image and general image). Figure should include all the hypotheses proposed.
I propose that both literature review and hypotheses section will have to go through some revision.
Research methods: I find that the choice of “destinations under investigation included: Muslim destinations (Egypt, Morocco and Indonesia) and Christian destinations (Spain, Brazil and Colombia)”, should be better justified. It should also consider whether this is a case study. For example, what is the difference between the economic development and security levels between these countries?
It is good to know where each item came from so can we please have a table in appendices with all the constructs and their items including their sources?
Can we also have a table with the results of the Exploratory Factor Analysis? It will help to see how the constructs were formulated and justify the use of the method.
Results: Section is good presenting important measurements and results: KMO, Bartlett’s test of sphericity, composite reliability (CR) and average variances extracted (AVE), Cronbach’s and the β’s.
Figure 2 will need proofreading. Some squares present different font and size, and some are missing information.
Discussion: Dear authors we will need a discussion section comparing your results with previous literature review. This can be done in a new section or embedded in the results section. We are missing a critical part of the outcomes of this research compared to previous knowledge and possible contributions.
Conclusion: Good section summarizing the research. Maybe there could be more practical implications to be discussed. I find that it is missing possible limitations and future research.
Comments on the Quality of English Language
I believe that the manuscript should be proofread before publication.
Author Response
We have conducted a thorough revision of the manuscript and made a point-to-point response to each of the comments raised. As a result, we believe the manuscript has greatly improved. We hope our revision meets with the requirements. Our comments in bold letters
Comments and Suggestions for Authors
Dear authors I found this research very interesting. The idea of investigating religion in perceived risk for destination image to be useful for both practical and theoretical implications. I found your manuscript interesting and engaging however I have made some notes:
Introduction: Section has good flow easy to follow. It presents the topic of discussion and the need for the research. I believe it would benefit from having the aim of this paper more clearly presented in the end and some expected theoretical and practical implications of this research.
Theoretical and practical implications have been added in the introduction
Literature review: Section has some very good arguments and adequate current relevant evidence to support them. However, it does not serve the purpose to define several important topics such as: perceived risk, unperceived risk, personal security, safety, terrorism, political turmoil strike. Furthermore, looking at the hypothesis chapter it does not define and discuss topics such as religion, cognitive image, affective image and general image.
We have defined the topics named by the reviewer and referenced them with the hypotheses.
Hypotheses: This section is a bit weak. It does state that “Based on the literature review presented above we formulate some hypotheses”, but it does not present any previous literature that examined the relationship between the proposed hypotheses.
We have included the hypotheses in the review section rather than in a separate section.
Figure 1: Figure only presents H1, H2 and H3 ignoring H1a, H2a and H3a. It also introduces new topics (cognitive image, affective image and general image). Figure should include all the hypotheses proposed.
All hypotheses have been included in Figure 1.
I propose that both literature review and hypotheses section will have to go through some revision.
We have revised and restructured the literature review and hypotheses sections.
Research methods: I find that the choice of “destinations under investigation included: Muslim destinations (Egypt, Morocco and Indonesia) and Christian destinations (Spain, Brazil and Colombia)”, should be better justified. It should also consider whether this is a case study. For example, what is the difference between the economic development and security levels between these countries?
We have better justified the choice of destinations in Section on 3.1 Study Site and Sampling
It is good to know where each item came from so can we please have a table in appendices with all the constructs and their items including their sources?
Can we also have a table with the results of the Exploratory Factor Analysis? It will help to see how the constructs were formulated and justify the use of the method.
We have added an Appendix with the results of the factor analysis of the items.
Results: Section is good presenting important measurements and results: KMO, Bartlett’s test of sphericity, composite reliability (CR) and average variances extracted (AVE), Cronbach’s and the β’s.
Figure 2 will need proofreading. Some squares present different font and size, and some are missing information.
Figure 2 corrected.
Discussion: Dear authors we will need a discussion section comparing your results with previous literature review. This can be done in a new section or embedded in the results section. We are missing a critical part of the outcomes of this research compared to previous knowledge and possible contributions.
Conclusion: Good section summarizing the research. Maybe there could be more practical implications to be discussed. I find that it is missing possible limitations and future research.
We have added Title 5 Discussion, 6. Conclusions, 6.1. Theoretical and Managerial Implications and 6.2. Limitations and Prospect Directions.
Comments on the Quality of English Language
I believe that the manuscript should be proofread before publication.
We would like to thank reviewers for the constructive and valuable comments.
Reviewer 2 Report
Comments and Suggestions for Authors
The title...The Influence of Muslim and Christian Destinations on Tourists' Behavioural Intentions and Risk Perceptions, is not very coherent with the study presented here. Despite the fact that this is a very solid work in methodological terms, and the analysis of the data, it is not clear how the religion of the tourist destination, in terms of perception, can be assessed using this methodology and in this study in particular.
The bibliographical references, although correct and well-framed, are somewhat dated.
Regarding the risk perception:
Reliability and validity of the measurement items for Risk perception
Item 5: Getting a disease transmitted by an animal...I don't think this is the right wording, as it could be taken to mean the transmission of diseases transmitted from the consumption of contaminated food, such as brucellosis or tuberculosis, or from animal attacks such as reptile poisoning or rabies. The diseases to which the authors refer - malaria, are infectious and transmitted by insect bites, and are therefore very specific to certain regions. That's why the wording of the question should be revised.
Regarding Item 6: To have a fire in the hotel...I think that an item should have been included regarding the effectiveness or existence of emergency services in the event of an accident or disaster.
There is a clear lack of reference in terms of risk perception descriptors in relation to the impact that epidemiological situations such as the pandemic can have on a tourist destination, i.e. what capacity a tourist destination has to deal with epidemic emergencies?
Author Response
We have conducted a thorough revision of the manuscript and made a point-to-point response to each of the comments raised. As a result, we believe the manuscript has greatly improved. We hope our revision meets with the requirements. Our comments in bold letters.
Comments and Suggestions for Authors
The title...The Influence of Muslim and Christian Destinations on Tourists' Behavioural Intentions and Risk Perceptions, is not very coherent with the study presented here. Despite the fact that this is a very solid work in methodological terms, and the analysis of the data, it is not clear how the religion of the tourist destination, in terms of perception, can be assessed using this methodology and in this study in particular.
It is now more clearly explained that multigroup analaysis makes a comparison within homogenous and between heterogenous groups, based on those characteristics that can differentiate groups such as gender, age, nationality, customs, culture and beliefs.
The bibliographical references, although correct and well-framed, are somewhat dated.
We have replaced the older bibliography with more recent ones.
Regarding the risk perception:
Reliability and validity of the measurement items for Risk perception
Item 5: Getting a disease transmitted by an animal...I don't think this is the right wording, as it could be taken to mean the transmission of diseases transmitted from the consumption of contaminated food, such as brucellosis or tuberculosis, or from animal attacks such as reptile poisoning or rabies. The diseases to which the authors refer - malaria, are infectious and transmitted by insect bites, and are therefore very specific to certain regions. That's why the wording of the question should be revised.
Regarding Item 6: To have a fire in the hotel...I think that an item should have been included regarding the effectiveness or existence of emergency services in the event of an accident or disaster.
Items 5 and 6 have been renamed more clearly and correctly.
There is a clear lack of reference in terms of risk perception descriptors in relation to the impact that epidemiological situations such as the pandemic can have on a tourist destination, i.e. what capacity a tourist destination has to deal with epidemic emergencies?
As discussed in the literature review, health risk is the most important risk in tourists' risk perception. However, as health risk is not the focus of our paper, we have not considered epidemiological risk.
We would like to thank reviewers for the constructive and valuable comments.
Reviewer 3 Report
Comments and Suggestions for Authors
I had the pleasure of reviewing the manuscript titled “The Influence of Muslim and Christian Destinations on Tourists´ Behavioral Intentions and Risk Perceptions” to be considered for publication in "Behavioral Sciences (ISSN 2076-328X)." The research seems sound and provides fairly interesting findings, yet it requires some substantial improvements. Specifics are below:
1- The title: the title is okay.
1- The abstract: abstract looks okay. It is possible to add the program that was used to analyze the data.
2- The introduction: The introduction needs some amendments.
· The introduction should revolve around the three research variables (Risk perception (and its dimensions), the image of the destination (and its dimensions), and the intentions to visit a destination) in addition to the moderating variable (Predominant religion).
· At the end of the introduction, some practical contributions to tourist destinations can be made.
· The introduction should provide reasonable justification for the study and reflect its significance. In other words, why this study is important? How it is different from previous studies? How can study contribute to literature?
3- Literature: the literature needs some amendments:
· A separate title must be specified for each hypothesis in which the variables and the relationship between them are explained.
· Although the study revolves around the moderating role of "predominant religion" in the tourist destination, no separate title has been allocated to this relationship, nor has it been studied in-depth in the study's theoretical framework.
· In Figure 1, only three hypotheses are indicated. The other hypotheses should be mentioned.
4- Methodology: seem to be thorough. Yet, it can be improved by addressing some points:
· The difference between the moderating effect and multi-group analysis tests must be ascertained.
· In Figure 2, there are unclear values.
· In line 322, the authors say that the destination plays the role of a moderator. Is it the destination or the religion?
5- Title 6:
· Title 6 should be divided into a discussion (in which the results are compared with previous literature), theoretical contributions, practical contributions, conclusions, and the limitations of the study and future studies.
Author Response
We have conducted a thorough revision of the manuscript and made a point-to-point response to each of the comments raised. As a result, we believe the manuscript has greatly improved. We hope our revision meets with the requirements. Our comments in bold letters.
Comments and Suggestions for Authors
I had the pleasure of reviewing the manuscript titled “The Influence of Muslim and Christian Destinations on Tourists´ Behavioral Intentions and Risk Perceptions” to be considered for publication in "Behavioral Sciences (ISSN 2076-328X)." The research seems sound and provides fairly interesting findings, yet it requires some substantial improvements. Specifics are below:
1- The title: the title is okay.
1- The abstract: abstract looks okay. It is possible to add the program that was used to analyze the data.
We place it in the section 3.2. Modelling, where we consider it more appropriate.
2- The introduction: The introduction needs some amendments.
- The introduction should revolve around the three research variables (Risk perception (and its dimensions), the image of the destination (and its dimensions), and the intentions to visit a destination) in addition to the moderating variable (Predominant religion).
Following the reviewer's recommendations, we have added and commented on the variables mentioned and the moderating variable in the introduction.
- At the end of the introduction, some practical contributions to tourist destinations can be made.
Practical contributions have been added in the Introduction section
- The introduction should provide reasonable justification for the study and reflect its significance. In other words, why this study is important? How it is different from previous studies? How can study contribute to literature?
We have added in the introduction section comments on the importance of this study, its contribution to the literature and its differences with previous studies.
3- Literature: the literature needs some amendments:
- A separate title must be specified for each hypothesis in which the variables and the relationship between them are explained.
We have added the hypotheses and explanation of the proposed causal relationships in the literature review section instead of in a separate section as we had done previously.
- Although the study revolves around the moderating role of "predominant religion" in the tourist destination, no separate title has been allocated to this relationship, nor has it been studied in-depth in the study's theoretical framework.
We have expanded the literature review on the potential influence of the predominant religion in tourist destinations, and added a Discussion section.
- In Figure 1, only three hypotheses are indicated. The other hypotheses should be mentioned.
All hypotheses have been included in Figure 1.
4- Methodology: seem to be thorough. Yet, it can be improved by addressing some points:
- The difference between the moderating effect and multi-group analysis tests must be ascertained.
We have added a more precise explanation of the multigroup analysis and the modelling effect in the section 4.3. Moderating Effects
- In Figure 2, there are unclear values.
Figure 2 corrected.
- In line 322, the authors say that the destination plays the role of a moderator. Is it the destination or the religion?
Where the line 322 was, we have written “predominant religion” to clarify that it is not destination but religion.
5- Title 6:
- Title 6 should be divided into a discussion (in which the results are compared with previous literature), theoretical contributions, practical contributions, conclusions, and the limitations of the study and future studies.
Title 6 is now Title 5 Discussion, 6. Conclusions, 6.1. Theoretical and Managerial Implications and 6.2. Limitations and Prospect Directions.
We would like to thank reviewers for the constructive and valuable comments.
Round 2
Reviewer 3 Report
Comments and Suggestions for Authors
I see that the authors have largely adhered to the required comments. This version is well improved .
Congratulations to the authors